# Monitoring of Phosphorus Compounds in the Influence Zone Affected by Nuclear Power Plant Water Discharge in the Styr River (Western Ukraine): Case Study

Pavlo Kuznietsov [1] , Olha Biedunkova [1] and Yuliia Trach [1,2,*]

1. Institute of Agroecology and Land Management, National University of Water and Environmental Engineering, 33028 Rivne, Ukraine; p.m.kuznietsov@nuwm.edu.ua (P.K.)
2. Faculty of Civil and Environmental Engineering, Institute of Civil Engineering, Warsaw University of Life Sciences, 02-787 Warsaw, Poland
* Correspondence: y.p.trach@nuwm.edu.ua

**Abstract:** The main causes of surface water pollution with phosphate ions are various human activities. Monitoring the content of phosphorus compounds in surface waters is important for the management of water bodies. Phosphorus is an essential element for the life of flora and fauna, but in excessive amounts it can have a harmful effect on the environment. The inflow of phosphorus compounds into the Styr River (Western Ukraine) occurs as a result of the discharge of cooling water from the Rivne NPP's cooling water system. This article has three purposes: (1) The inflow of phosphorus compounds to the Styr River occurs with the discharge of cooling water using 1-hydroxyethylidene-1,1-diphosphonic acid (HEDP); (2) phosphorus compounds (phosphate ions, HEDP, and total phosphorus in surface waters of the Styr River) are monitored and analyzed, and the analysis of the quality of river water is carried out in accordance with environmental standards for the content of phosphorus compounds in the zone of influence of the Rivne NPP; (3) in terms of phosphorus content, the quality of the water of the Styr River, after the discharge of the Rivne NPP's cooling water, is characterized as "satisfactory" and belongs to Class III. A seasonal trend of changes in the content of phosphate ions and total phosphorus was found, and the concentration of HEDP in the water of the Styr River depends on the technological dosage mode during the corrective treatment of the Rivne NPP.

**Keywords:** nuclear power; phosphates; 1-hydroxy ethylidene-1,1-diphosphonic acid; ecosystem; return water; discharge

## 1. Introduction

Sustainability and phosphorus compounds in river water are important topics related to environmental conservation and the responsible management of natural resources [1]. Efforts to monitor phosphorus compounds in river water can be part of a sustainable solution that can help minimize environmental impacts [2]. Addressing phosphorus issues in river water often requires monitoring its supply with water discharge from industrial anthropogenic sources [3]. In summary, the management of phosphorus compounds in river water is essential for sustainability [4]. It involves addressing sources of pollution, implementing sustainable practices, and working together to influence technological and anthropogenic factors [5].

Atomic energy is a safe and stable source of electricity production. For the operation of an atomic power plant (NPP), it is necessary to ensure cooling of its elements and systems, which are powered by large amounts of water that then discharge into a water body [6,7]. In the cooling systems of nuclear power plants, heating, evaporation, and concentration of the cooling water components occur [8,9]. To ensure the water–chemical regime, nuclear power plants use corrective treatment with phosphorus-containing reagents [10]. According to the

existing regulations [11], the discharge limits for phosphorus compounds from different NPPs range from 0.6 to 100 t/year; significant discharge limits for phosphorus compounds for NPPs are primarily due to the large amount of water used for cooling.

Water is an important resource for the functioning of any type of industry. Rapid urbanization, industrialization, and intensification of industrial production have led to a deterioration of surface water quality worldwide [12–14]. A high level of pollution entering the environment can make water unsuitable for use, and an excess of nutrient biogenic elements can cause eutrophication of water bodies and even the death of fish [15,16]. Phosphorus is an important biogenic element, but due to the significant volumes of its anthropogenic input, the natural phosphorus cycle has undergone changes in recent years. The greatest environmental load from anthropogenic phosphorus input is observed in rivers, which is why the reduction of the environmental risk of river pollution with phosphorus compounds is an actual problem [17].

Phosphorus flows into rivers from point sources and diffuse sources, which may contain organic and inorganic forms of this element [18,19]. Diffuse sources are contributions from the leaching of geological rocks and land use, while point sources include industrial discharge. In surface waters, the most common forms of phosphorus are its inorganic compounds in the form of phosphate and polyphosphate ($PO_4^{3-}$), while organic phosphorus is encountered as a result of the life activities and decomposition of aquatic organisms, as well as anthropogenic factors that contribute to phosphorus discharge [20,21].

Inorganic forms of phosphorus compounds in natural waters are represented by 42% compounds with magnesium ($MgPO_4^-$), 29% hydrogen phosphate ions ($HPO_4^{2-}$), 15% sodium hydrogen phosphates ($NaHPO_4^-$), and 12% hydrogen phosphates and calcium phosphate ($CaHPO_4$, $CaPO_4^-$) [22].

Organic forms of phosphorus compounds in natural waters include phosphonates, myo-inositol hexakisphosphates, and diesters of orthophosphate. Phosphonates exhibit processes of biodegradation, complexation, and sediment adsorption in the surrounding environment due to their structural similarity to phosphate esters. Phosphonates often act as enzyme inhibitors [23]. Up to 80% of soluble phosphonates undergo complete biodegradation in surface waters through the cleavage of the C-P bond [24], with the end products of decomposition being hydrocarbons and phosphate ions. The most common anthropogenic phosphorus compounds entering aquatic ecosystems are phosphonate in hydrocarbons and phosphate ions. The most widely used phosphonate in industry is 1-hydroxyethylidene-1,1-diphosphonic acid (HEDP) [25].

Organic and inorganic forms of phosphorus compounds in surface waters can transform into each other. Phytoplankton cells absorb phosphates released during the oxidation of accumulated organic phosphorus [26]. It is interesting that in different parts of the globe, in various types of ecosystems, the phosphate content in water has diverse effects on phytoplankton parameters. Thus, in the coastal waters of India, phosphorus has less of an effect on phytoplankton diversity compared to nitrate content in water [27]. Similar results were obtained in the coastal waters of China, where the number and species diversity of phytoplankton depend on the phosphate content in the water much less, compared to the temperature factor and the nitrate content [28]. The study of the river–lake system revealed that for the number and diversity of lake phytoplankton, the C:N ratio is an important regulatory factor, while in rivers the C:N:P ratio is the main factor [29]. In surface waters, inorganic forms of phosphorus compounds account for an average of up to 50% of the total amount, but the ratio between organic and inorganic phosphorus can vary widely [30].

In small rivers in Ukraine (river length up to 100 km), the concentration of inorganic phosphorus forms varies from trace concentrations to 0.7 mg/dm$^3$, while in medium (river length from 100 km to 500 km) and large rivers (river length over 500 km), it ranges from 0.07 to 0.50 mg/dm$^3$ [31]. The lowest phosphorus content (0.02–0.1 mg/dm$^3$) is observed in August–September. Particularly high concentrations were recorded in some rivers of the eastern region of Ukraine, reaching [32]. Due to natural mechanisms of phosphonate elimination, the prolonged release of bioavailable phosphate from phosphonates into

natural waters is ensured, thus not excluding eutrophication processes [33]. The regulation of phosphorus discharge into wastewater in Ukraine is carried out in accordance with [34], while in EU countries, phosphorus content is regulated according to [35].

In general, analysis of current river water quality studies shows that nutrient input to rivers is significantly negative due to climate change [36–38]. An important tool for solving this problem is forecasting water quality, including possible scenarios of phosphate content changes [39,40]. However, numerous biological, hydrological, and other factors in river ecosystems complicate prediction [41]. Therefore, modern scientists often use simple mathematical models, such as coring, to analyze the formation of the regime of nutrients in river water. Such approaches allow the possibility of recording which natural or anthropogenic factors are the most important causes of the presence of nutrients in river water [42].

Given the importance of ensuring the ecological safety of surface waters, there is great interest in determining anthropogenic pollution of water bodies with phosphorus. The purpose of our research was to assess the pollution of the Styr River by phosphorus compounds originating from the return waters of the Rivne Nuclear Power Plant (Rivne NPP) as a result of anti-scale correction of phosphonate treatment. The information and analysis of such monitoring are important for planning and implementing the necessary water resource management strategies. Considering the specifics of the operation of the nuclear power plant (the formation of large volumes of return water), it is necessary to constantly focus on the quality of surface water in order to prevent water pollution in reservoirs and the deterioration of the environment in general.

## 2. Materials and Methods

The research was conducted on the process water of the Rivne Nuclear Power Plant (Rivne NPP) and the surface water of the Styr River. The Rivne NPP consists of four VVER-type nuclear reactors and is located in Ukraine, Eastern Europe. The cooling system of the NPP is an open-type cooling system with water cooling in cooling towers (Figure 1). The basin scheme and the impact zone of the Rivne NPP discharge are indicated in Figure 1. The hot cooling water flows from the condenser through the cooling tower where it is cooled by evaporation, and the cold cooling water is returned through the process and reheated. The CCS cooling water concentration cycles are controlled by blowdown, which is sent to a reservoir; water losses due to evaporation and blowdown are replenished with additional water taken from the reservoir (Figure 1). Corrective reagents are used for water treatment, in particular phosphonate treatment with HEDP implemented at the Rivne NPP [33].

The cooling water consumption for power units No. 1 and 2 (VVER-440) is 91,000 $m^3$/hour for each power unit, while for power units No. 3 and 4 (VVER-1000), it is 188,900 $m^3$/hour each. The intake and discharge of cooling water at the Rivne NPP are carried out in the Styr River. The Styr River is not navigable, with a channel width of 40–60 m, a depth ranging from 0.8 to 2 m, and a flow velocity of 0.4–0.7 m per second. The dynamics of water consumption from the Styr River for the Rivne NPP range from 50 to 73 million $m^3$/year, with discharge ranging from 12 to 18 million $m^3$/year, and it does not exceed the established water use limits. Since 2015, due to the established aeration and nourishment regime, there has been a tendency to reduce water intake and discharge from the Rivne NPP (Figure 2).

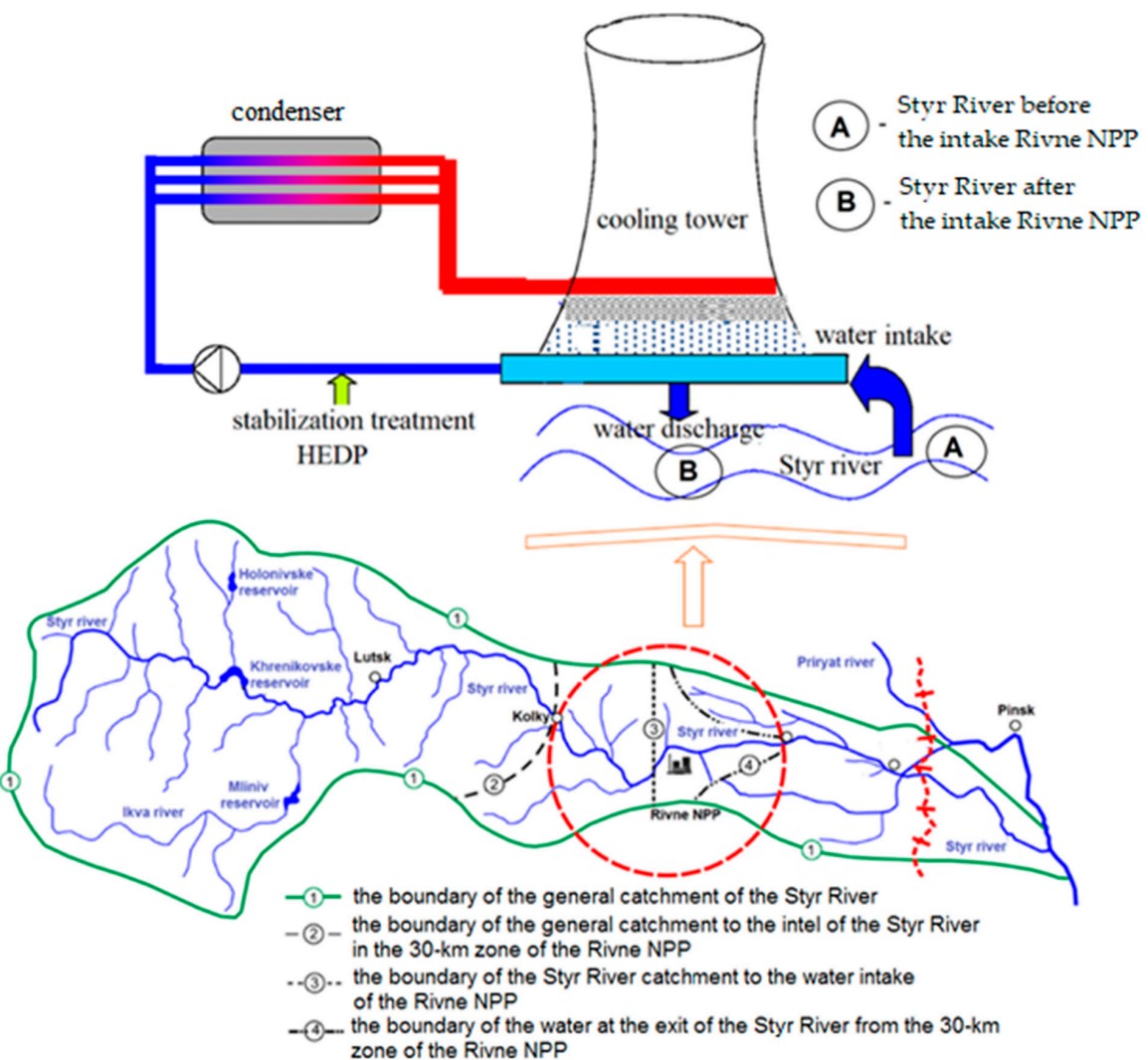

**Figure 1.** Schematic diagram of the cooling system at the Rivne NPP and Rivne NPP impact zones.

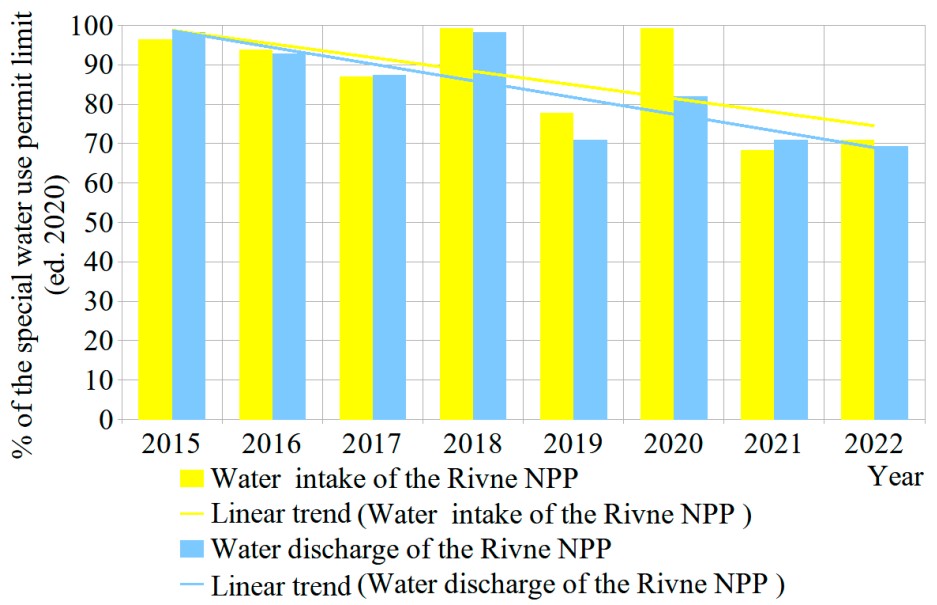

**Figure 2.** Dynamics of water use at the Rivne NPP's CCS.

The hydrochemical composition of the Styr River is formed under conditions of excessive moisture and the influence of widespread carbonate rocks. The river is also fed by artesian waters from a karstified limestone layer, which leads to an increased concentration of calcium ions ($Ca^{2+}$) and bicarbonate ions ($HCO_3^-$) in the river water. Eutrophication is observed in the Styr River in the warm season on sections of the shoreline with slow flow (Figure 3).

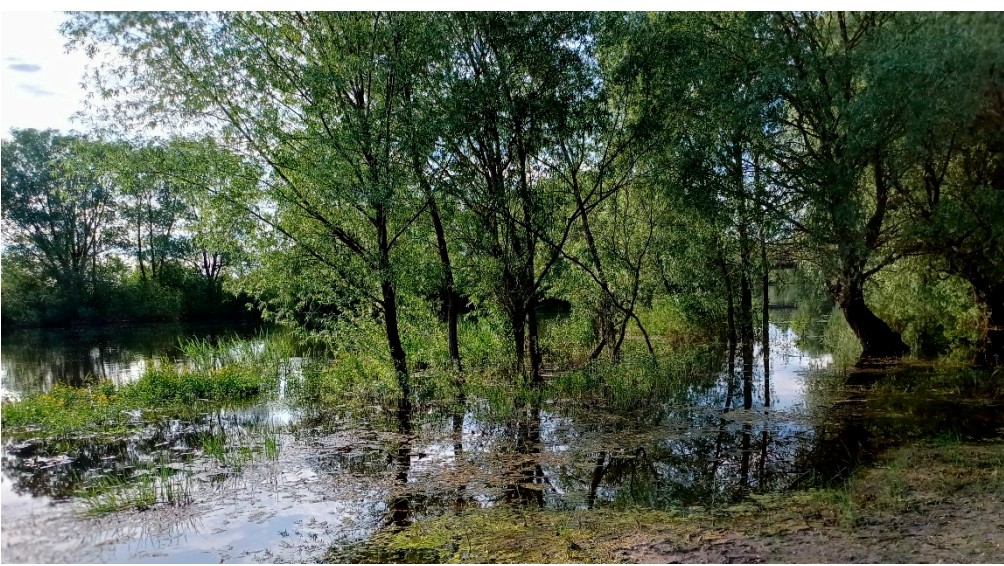

**Figure 3.** Eutrophication phenomenon with algae and vegetation growth in the Styr River (May 2023).

The assessment of the quality of the Styr River water based on the content of phosphorus compounds was carried out according to the existing regulations, and the classification of surface waters in Ukraine and the European Union according to [43–47] are listed in Table 1. The assessment of water quality included environmental assessment (EA), hygienic assessment (HA), and water management assessment (WMA) for both domestic (DW) and fisheries (FW) water use. The assessment of water quality was performed using a detailed analysis method, which involved comparing the measured value of the parameter with its standard. Water qualities corresponding to Classes I, II, and III were characterized as "excellent", "clean", and "satisfactory", respectively (Methodology, 2019). Within these classes, the values of various forms of phosphorus indicated low levels of anthropogenic influence and deviated only slightly from the values typical for the mass of surface water under reference conditions; concentrations of chemical and physico-chemical parameters did not exceed environmental quality standards (Table 1). The methods [34,48] were used to determine the ecological status of the Styr River in the area of influence of the Rivne NPP discharge with the calculation using the software Microsoft Office Excel 2019 "River Phosphorus Calculator" (http://www.wfduk.org/resources/rivers-phosphorus-standards, accessed on 1 October 2023).

In the assessment of water quality, the results of monitoring from accredited laboratories at the Rivne NPP (Certificate of Recognition of Measurement Capabilities No. R-8/11-57-5 dated 22 December 2017) were used. The measurements were carried out using standardized methods [49,50] with the following metrological characteristics (Table 1), which were verified under the State Metrological Supervision of Ukraine.

**Table 1.** Standards for surface water quality assessment and measurement methods used in the study of phosphorus compound content.

| Parameter | Ecological Indicator | | | | | | Measurement Methods | |
|---|---|---|---|---|---|---|---|---|
| | EA | Ukrainian | | | EC | | $C_{min}$–$C_{max}$ [(3)], mg/dm$^3$ | $\delta$ [(4)], % |
| | | HA | WMA | | HA | FW | | |
| | | | DW | FW | | | | |
| Phosphates, mg PO$_4^{3-}$/dm$^3$ | <0.015–0.10 mg P/dm$^3$ (I, II, III Classes) | 3.5 | 3.5 | 2.14 | - | 0.2–0.4 [(2)] | 0.05–100 | ± 15 [(5)] |
| Pure phosphorus, mg P/dm$^3$ | | - | - | 0.7 | - | - | | ± 10 [(5)] |
| HEDP 1, mg/dm$^3$ | - | 0.3 [(1)] | 0.3 [(1)] | 0.9 | - | - | 0.06–4.0 | ± 18 |

Notes: [(1)]—indicated minimum value for sodium salt; [(2)]—depending on the assignment; [(3)]—$C_{min}$–$C_{max}$—measurement range; [(4)] $\delta$—relative measurement error; [(5)]—from 0.05 to 0.5 inclusive, mg/dm$^3$, $\delta$ = ±15%, above 0.5 to 100 inclusive, mg/dm$^3$ $\delta$ = ±10%.

The spectrophotometric measurement method for phosphate concentration employed in the study is based on the reaction of phosphorus with ammonium molybdate $((NH_4)_6Mo_7O_{24} \times 4H_2O)$ in the presence of stannous chloride $(SbCl_3 \times 6H_2O)$, resulting in the formation of a blue-colored compound.

The measurement of HEDP concentration used in the study is based on the breakdown of the P-C bond of phosphonates using potassium persulfate and the conversion of phosphonates into phosphates, followed by spectrophotometric measurement of the total phosphorus concentration, determined as the sum of HEDP and phosphate ions, measured using the measurement method (Table 1).

The assessment of the biogenic elements was carried out according to the methodology [41], using C:P:N stoichemistry. Data analysis was performed using standard methods of mathematical statistics with the assistance of software [51]. A density estimation of the data series was conducted using the method of [52], and the seasonality of changes was determined using the Probabilistic Neural Network (PNN) Classifier, which is applied to identify the distribution pattern of variables in each group and assess the density function of each group to determine classification characteristics for observation groups [53].

## 3. Results

### 3.1. Estimation of Phosphorus Input from Discharge

The water intake of the Rivne Nuclear Power Plant (NPP) from the Styr River should not exceed 74 million m$^3$/year under the established conditions [50]. The actual average water intake is 50 million m$^3$/year.

The discharge of technological water from the Rivne NPP into the Styr River should not exceed 14 million m$^3$/year under the conditions [54], and the actual average discharge of return water is 11 million m$^3$/year (Figure 4). The maximum allowable discharge of phosphate ions for the Rivne NPP into the Styr River is set at 57.44 tons/year, and for HEDP, it is 16.57 tons/year.

The actual discharge values of phosphate ions into the surface water of the Styr River for the Rivne NPP ranged from 3.5 to 4.5 tons/year, with a discharge limit of 57.44 tons/year. For HEDP, the range of actual discharge ranged from 5.3 to 15.8 tons/year, with a discharge limit of 16.57 tons/year (Figure 4).

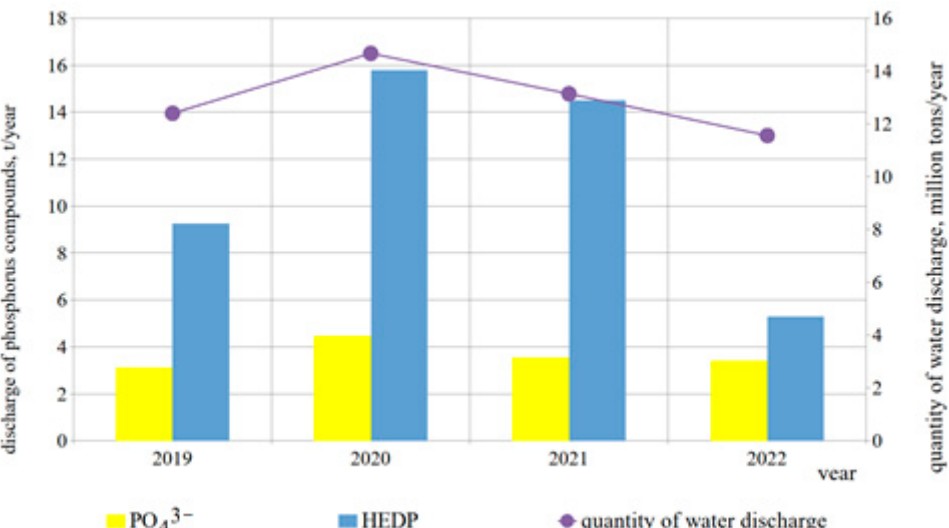

**Figure 4.** Actual quantities of phosphorus compounds discharged into the surface water of the Styr River by the Rivne NPP.

The actual discharge of phosphorus compounds into the water of the Styr River for the Rivne NPP does not exceed the prescribed discharge limits.

### 3.2. Results of Chemical Control of Phosphorus Compounds

In accordance with the applied control methods, inorganic forms of phosphorus (phosphates and polyphosphates) were measured. The results of chemical control were presented in terms of ($PO_4^{3-}$) and organic forms of phosphorus (HEDP) in terms of HEDP. The mass concentration of total phosphorus was determined as the sum of organic and inorganic forms of phosphorus in terms of phosphorus (P). The mean concentration of phosphate ions with standard deviation (M ± m) in the water of the Styr River before the water intake by the Rivne NPP was $0.275 \pm 0.118$ mg$PO_4^{3-}$/dm$^3$ and varied in the range (min–max) from $0.070$ mg$PO_4^{3-}$/dm$^3$ to $0.535$ mg$PO_4^{3-}$/dm$^3$. After the discharge of Rivne NPP's CCS wastewater, the mean concentration of phosphate ions was $0.296 \pm 0.141$ mg$PO_4^{3-}$/dm$^3$ and varied in the range from $0.080$ mg$PO_4^{3-}$/dm$^3$ to $0.590$ mg$PO_4^{3-}$/dm$^3$ (Figure 5).

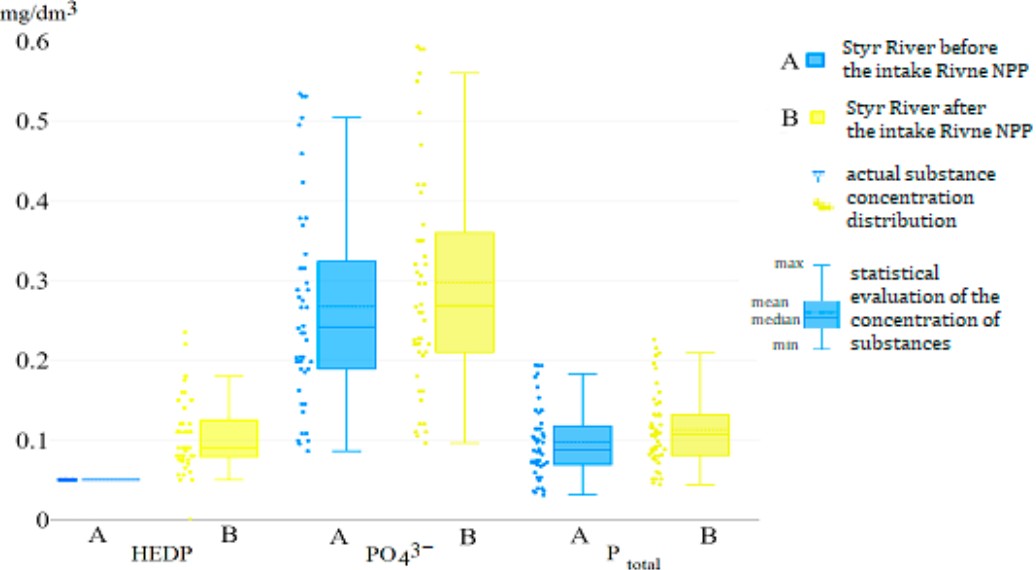

**Figure 5.** Concentration of total phosphorus, HEDP, and phosphate ions in the water of the Styr River before the intake and after the discharge of return water by the Rivne NPP (2019–2022).

The mean concentration of HEDP in the water of the Styr River upstream of the Rivne NPP intake is not determined by the applied measuring method (Table 1). During the period of the study it was below the lower measuring range (0.05 mgHEDP/dm$^3$) and did not change. The mean concentration of HEDP in the water of the Styr River after the discharge of the Rivne NPP's CCS effluent was (M ± m) 0.12 ± 0.09 mgHEDP/dm$^3$ and varied in the range (min–max) from 0.05 mgHEDP/dm$^3$ to 0.24 mgHEDP/dm$^3$ (Figure 5). The mean concentration of the total phosphorus in the water of the Styr River before the Rivne NPP water intake was 0.104 ± 0.044 mgP/dm$^3$ and varied in the range (min–max) from 0.015 mgP/dm$^3$ to 0.198 mgP/dm$^3$. The mean concentration after the discharge of the Rivne NPP's CCS wastewater was (M ± m) 0.113 ± 0.052 mgP/dm$^3$ and varied in the range (min–max) from 0.015 mgP/dm$^3$ to 0.226 mgP/dm$^3$ (Figure 5).

The dynamics of changes in the concentration of phosphorus compounds in the water of the Styr River is characterized by the coefficient of variation (CV). Its value is in the range from 18.3 to 40.2 (Table 2). The monitoring of water quality by the content of phosphorus compounds in the water of the Styr River before the water intake and after the discharge of the effluent of the Rivne NPP's CCS revealed an increase in their concentrations. This was due to the discharge into the river of wastewater from the Rivne NPP with phosphate ions.

**Table 2.** Descriptive statistics of the content of phosphorus compounds in the water of the Styr River after the discharge of reverse waters from the Rivne NPP.

| Chemical Parameters | Parameter [1] | Year | | | |
|---|---|---|---|---|---|
| | | **2019** | **2020** | **2021** | **2022** |
| Phosphates, mg/dm$^3$ | M | 0.264 | 0.303 | 0.291 | 0.327 |
| | ±m | 0.096 | 0.167 | 0.131 | 0.170 |
| | min | 0.080 | 0.120 | 0.105 | 0.110 |
| | max | 0.420 | 0.495 | 0.590 | 0.590 |
| | Cv | 35.33 | 34.55 | 35.86 | 38.60 |
| HEDP, mg/dm$^3$ | M | 0.07 | 0.12 | 0.13 | 0.17 |
| | ±m | 0.03 | 0.08 | 0.15 | 0.10 |
| | min | 0.05 | 0.08 | 0.08 | 0.07 |
| | max | 0.09 | 0.18 | 0.24 | 0.15 |
| | Cv | 18.50 | 30.51 | 40.20 | 33.30 |
| Total Phosphorus, mg/dm$^3$ | M | 0.097 | 0.118 | 0.116 | 0.120 |
| | ±m | 0.030 | 0.050 | 0.040 | 0.050 |
| | min | 0015 | 0.051 | 0.051 | 0.046 |
| | max | 0.145 | 0.209 | 0.226 | 0.215 |
| | Cv | 18.30 | 30.32 | 35.23 | 34.76 |

Note: [1] M—arithmetic mean of the results; ±m—standard error of deviation; min, max—minimum and maximum values in the sample; Cv—coefficient of variation.

### 3.3. Form Distribution and Seasonal Variability of Phosphorus Content

The arithmetic mean concentration of organic forms of phosphorus in the surface water of the Styr River after the discharge of the Rivne NPP wastewater in terms of phosphorus molar concentration in 2019–2022 was 0.52 µmol/dm$^3$ with a trend of time changes from 0.842 µmol/dm$^3$ to 7.02 µmol/dm$^3$. Organic forms contribute up to 14% of the total phosphorus compound content, while inorganic forms make up 86% (Figure 6).

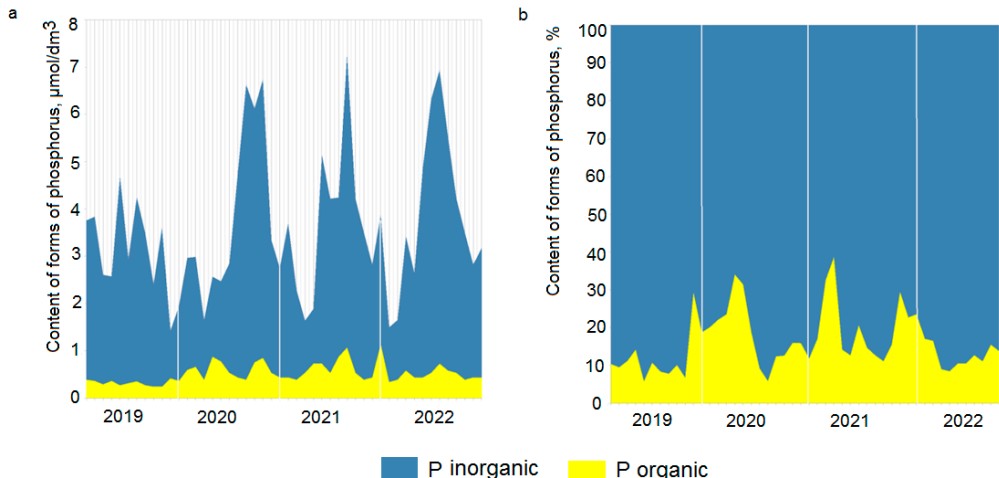

**Figure 6.** Distribution of the content of organic (P-organic) and inorganic (P-inorganic) phosphorus forms in the water of the Styr River within the influence zone of the Rivne NPP for the years 2019–2022 (**a**)—molar concentration, (**b**)—% content.

Seasonal fluctuations in the content of phosphate ions and total phosphorus in the water of the Styr River show an increasing tendency in the warm season (Figure 7). This is probably caused by the accumulation of phosphorus in the water during the intensification of biological processes in the warm season. The dynamics of HEDP content do not depend on the season and are determined by the technological dosing modes of phosphonate into the cooling water (Figure 7).

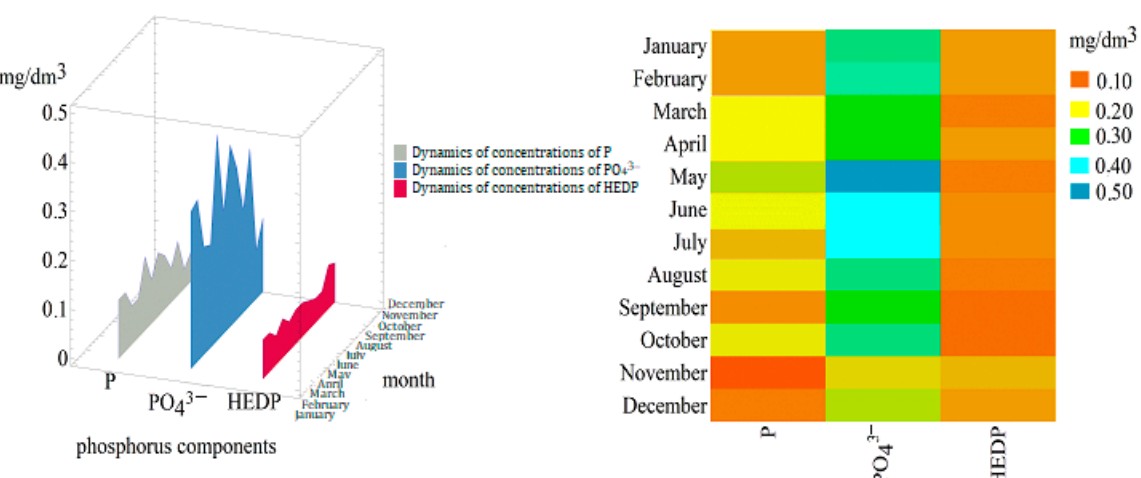

**Figure 7.** Seasonal fluctuations in the content of phosphorus forms in the water of the Styr River within the influence zone of the Rivne NPP for the years 2019–2022.

The analysis of seasonal variability in the content of phosphorus forms was conducted using the Probabilistic Neural Network (PNN) Classifier (Figure 8). As a result of selecting the number of network layers and neurons, an optimal model was chosen with a network structure consisting of two input parameters corresponding to organic and inorganic forms of phosphorus (Figures 6 and 7). The first layer of the classifier consists of 48 cases, the second layer has 13 neurons, and the third layer is the output layer classified according to probabilistic membership into 10 categories, four of which can be classified as annual seasonal fluctuations in phosphate ion data (Figure 8). It can be concluded that the dynamics of HEDP content do not depend on the season and are due to the technological modes of phosphonate dosing into Rivne NPP cooling water, in particular, the variability of HEDP dosing.

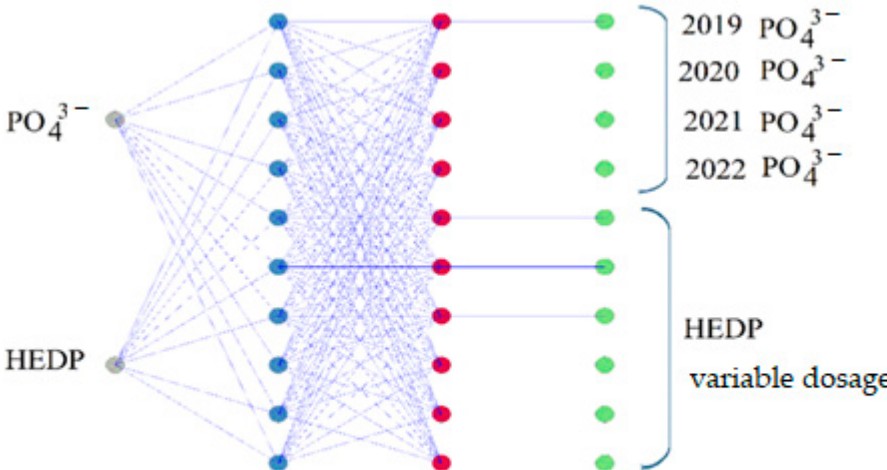

**Figure 8.** Detection of seasonal variations in phosphorus in the Styr River water using the PNN classifier.

Thus, an increase in the content of phosphorus compounds in the warm season may be associated with the development and death of a significant amount of phytoplankton, and minimal indicators of phosphorus compounds in the cold season with a decrease in phytoplankton life processes, as well as with anthropogenic pressure from agricultural activities with the application of phosphorus-containing fertilizers [55]. Also, a large amount of mobile phosphorus becomes gross (immobile) and settles to the bottom of water bodies, and the increase in phosphate content in the warm season may also be associated with the release of phosphorus from bottom sediments and decomposed phytoplankton and higher aquatic plant components [56].

### 3.4. Correlation of the Content of Phosphorus Compounds

The assessment of row density using the Multivariate Kernel Density Estimation method to detect correlations in the content of phosphorus forms is shown in Figure 9. The correlation between the content values of total organic and inorganic phosphorus forms is positive, with moderate significance between P/HEDP ($r = 0.25$) and $PO_4^{3-}$/HEDP ($r = 0.20$), and high between $P/PO_4^{3-}$ ($r = 0.89$).

The relationship between total phosphorus (P) and its inorganic ($PO_4^{3-}$) and organic (HEDP) forms has a linear correlation, statistically significant ($p = 0.0011$) at the high level ($r = 0.81$), with a linear dependence defined by the equation in Figure 9. The dependence equation, which determines the content of total phosphorus (P) as a function of its inorganic ($PO_4^{3-}$) and organic (HEDP) forms, can be used to follow the possible dynamics of changes in the balance of phosphorus forms due to the influence of natural or anthropogenic factors, or to predict the distribution of forms. The correlation of the values in the Styr River water between the increase in total phosphorus ($P_{total}$(B-A) before/after the Rivne NPP discharge and the total phosphorus ($P_{total}$(A) before the Rivne NPP water intake was established (Figure 10). The dependence, which determines the increase of phosphorus content depending on the phosphorus content before the water intake, has a linear inverse correlation, statistically significant ($p = 0.009$) at the average level ($r = 0.65$) with the dependence determined by the equation in Figure 10. The equation can be used to predict phosphorus discharge from the CCS at the Rivne NPP in order to optimize the technological processes of HEDP anti-scale treatment.

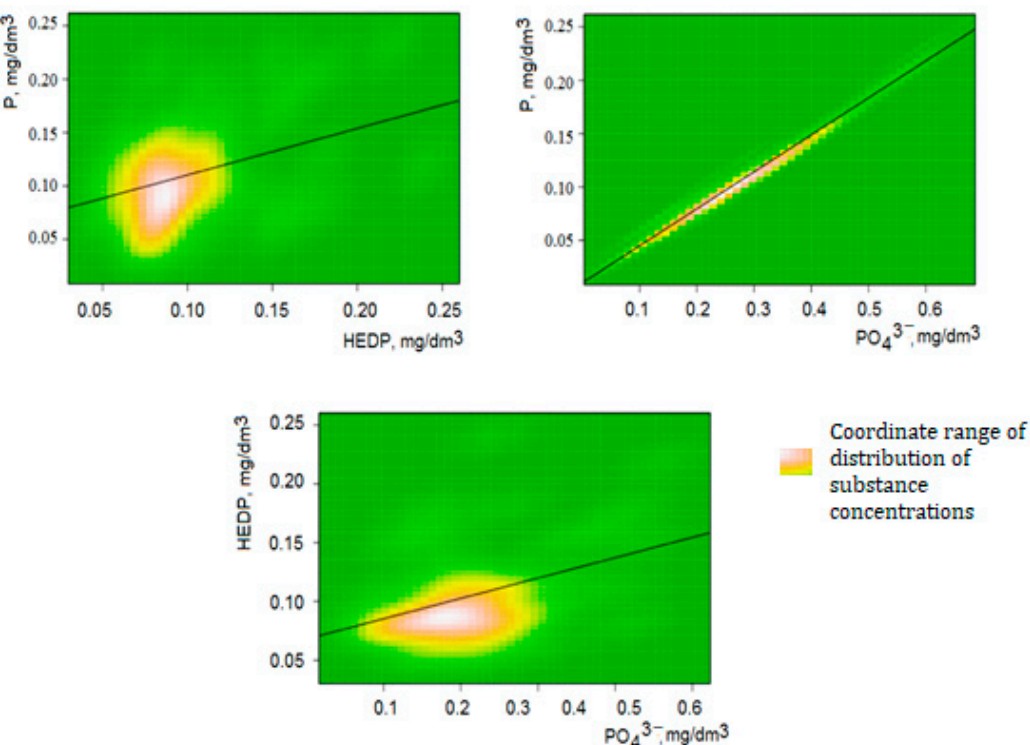

**Figure 9.** Multivariate Kernel Density Estimation for phosphorus compound content in the water of the Styr River in the Rivne NPP impact zone.

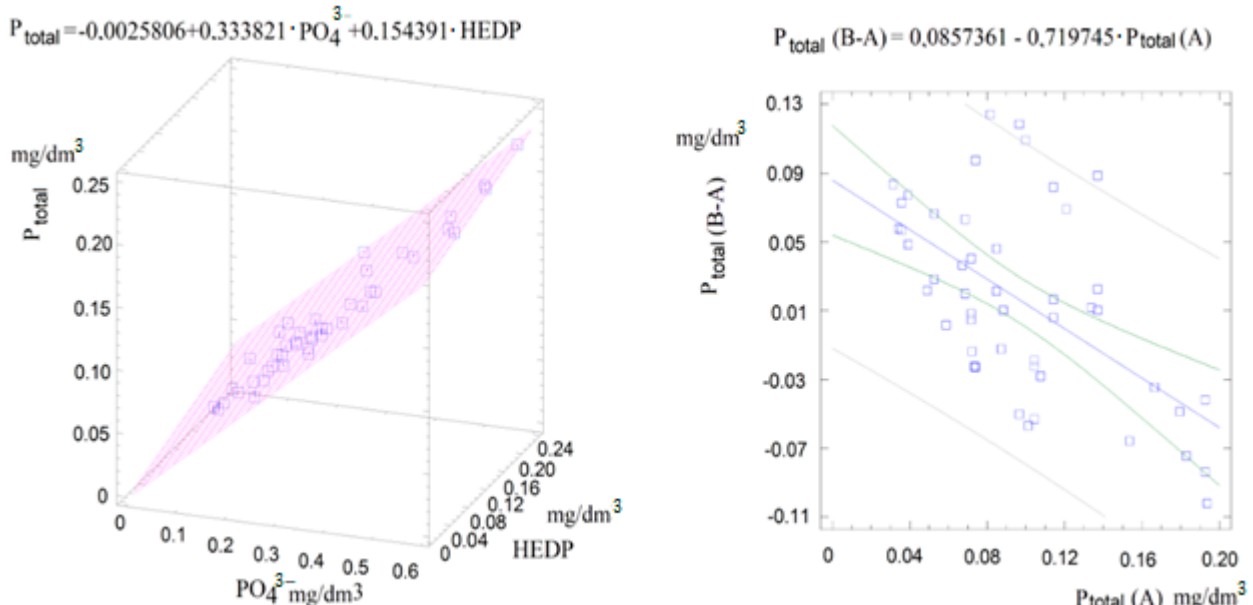

**Figure 10.** Correlation between the content of phosphorus compounds in the Styr River water and the increase in the content of total phosphorus.

### 3.5. Environmental Assessment of Water Quality by Phosphorus Content

The concentration of phosphorus compounds in the water of the Styr River does not exceed the maximum permissible concentration for fishing and domestic use according to Ukrainian standards, but in spring and summer it exceeds the established EU requirements for the quality of fresh water that needs protection or improvement to support fish life (Tables 1 and 2 and Figure 5).

The assessment of the ecological status by the phosphorus content of the Styr River water in the area of influence of the Rivne NPP effluent was carried out for samples before and after the discharge of the Rivne NPP effluent in accordance with the quality criteria applicable in Ukraine [34]. The P compliance with the Water Framework Directive [48], which takes into account or does not take into account the total alkalinity of the water (Table 3), was carried out based on of the average monthly monitoring results (Table 3), since the values have seasonal variability. Water with high alkalinity values can increase [57] the concentration of "Preference", which defines the limits of each ecological status, effectively providing weakening of the ecological standard. The extent of this effect will depend largely on the rivers fed by the limestone aquifer, rather than on rivers fed by groundwater with low alkalinity. The Styr is fed by carbonate limestone rocks, which is why the water has high alkalinity values (more than $mgCaCO_3/dm^3$), according to the environmental assessment of the ecological status of the Styr River. The Styr (Table 3) with alkalinity values taken into account has better values than those without alkalinity.

**Table 3.** Ecological assessment of phosphorus compounds in Styr River water before/after discharge of the Rivne NPP effluent for 2019–2022 according to the quality criteria of Ukraine (without alkalinity) and Water Framework Directive P compliance (with alkalinity).

| Month for 2019–2022 | Water from the Styr River to the Rivne NPP Water Intake | | River Styr after Discharge of Return Water from the Rivne NPP | |
| --- | --- | --- | --- | --- |
| | Environmental Threshold for Phosphorus without Alkalinity | Environmental Threshold for Phosphorus with Alkalinity | Environmental Threshold for Phosphorus without Alkalinity | With Alkalinity |
| January | III «satisfactory, contaminated» | «good» | III «satisfactory, contaminated» | «good» |
| February | III «satisfactory, contaminated» | «good» | III «satisfactory, contaminated» | «good» |
| March | IV «bad, dirty» | «moderate» | IV «bad, dirty» | «moderate» |
| April | IV «bad, dirty» | «moderate» | IV «bad, dirty» | «moderate» |
| May | V «very bad, very dirty» | «moderate» | V «very bad, very dirty» | «moderate» |
| June | IV «bad, dirty» | «moderate» | IV «bad, dirty» | «moderate» |
| July | IV «bad, dirty» | «moderate» | IV «bad, dirty» | «moderate» |
| August | III «satisfactory, contaminated» | «good» | III «satisfactory, contaminated» | «good» |
| September | III «satisfactory, contaminated» | «good» | III «satisfactory, contaminated» | «good» |
| October | III «satisfactory, contaminated» | «good» | III «satisfactory, contaminated» | «good» |
| November | II «good, clean» | «good» | II «good, clean» | «good» |
| December | III «satisfactory, contaminated» | «good» | III «satisfactory, contaminated» | «good» |

The ecological status of the Styr River water before and after the Rivne NPP effluent discharge is the same according to different methodologies, and no impact of the Rivne NPP effluent discharge on the Styr River water quality is observed. According to the quality criteria of Ukraine [34], the water of the Styr River before and after the Rivne NPP effluent discharge is characterized by a large change in the ecological status from II "good, clean" to V "very bad, very dirty", with a higher class and a tendency for pollution in spring and summer, which may be due to agronomic activities in the coastal zones with the application of phosphate fertilizers and intensification of natural biochemical processes in the warm season. According to the compliance criteria of the Water Framework Directive [48],

the water of the Styr River before and after the discharge of Rivne NPP wastewater is characterized by two classes of ecological status: 'moderate' in the spring and summer and 'good' for the rest of the year. This ecological status does not require improvement measures, taking into account the current value of water alkalinity [56,57].

The C:P:N stoichemistry in 2019–2021 was on the border of the Redfield ratio triangle with a slight shift in the balance towards an excess of carbon and a slight annual increase in the nitrogen contribution in 2022 (Figure 11); no excess or accumulation of phosphorus in the balance of biogenic elements C:N:P in the water of the Styr River before/after the discharge of the Rivne NPP wastewater was recorded. The C:P:N stoichemistry in water before/after the discharge from the NPP is comparable (Figure 11); there were no changes in the balance of biogenic elements of the ratio C:N:P in terms of their total content in 2019–2022 in the water of the Styr River before/after the discharge of the Rivne NPP wastewater.

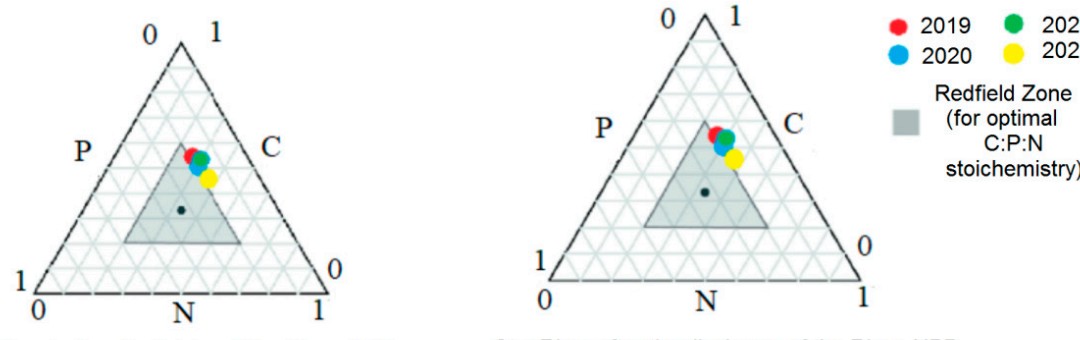

**Figure 11.** Assessment of the balance of biogenic elements according to C:P:N stoichemistry.

There is an excess of carbon in C:P:N stoichemistry, which is due to the formation of the hydrological regime of the Styr River with the feeding of castor limestone. In the future, based on the balance of nutrients, it is necessary to manage and control the supply of phosphorus, and the rather high values of HEDP discharge require the implementation of measures to minimize its use in the Rivne NPP process cycle.

## 4. Conclusions

Analytical and experimental studies have shown that the actual values of phosphate ions and HEDP discharge for the Rivne NPP do not exceed the permissible values. The maximum actual discharge value (2019–2022) of phosphate ions was up to 10% of the established discharge limit and HEDP up to 95% of the established discharge limit for water bodies used for fisheries and domestic purposes in Ukraine. However, it exceeded the EU standards, and did not exclude the possibility of eutrophication as "satisfactory", belonging to Class III. The concentration of phosphorus compounds in the water of the Styr River within the impact zone of the NPP showed seasonal dynamics with a tendency to increase during the warm period of the year. The analysis of phosphorus form dynamics using the PNN Classifier confirms the seasonal variability of inorganic forms of phosphorus concerning phosphate ions. A high, statistically significant correlation between the content of total phosphorus and phosphate ions was established. A linear equation describing the content of the total phosphorus, determined by both inorganic and organic forms, was calculated and can be used for further monitoring of possible phosphorus cycling changes influenced by the discharge from the Rivne NPP.

Overall, the research results indicate the absence of a negative impact of phosphorus compounds in the reverse water from the Rivne NPP on the ecosystem of the Styr River. However, the relatively high levels of HEDP discharge necessitate the implementation of measures to minimize its use in the technological cycle of the NPP and the continuation of water quality monitoring of the Styr river. It will be a perspective direction for further research regarding the sustainable use of water resources in the region.

**Author Contributions:** Conceptualization, P.K., O.B. and Y.T.; methodology, P.K. and O.B.; software, P.K.; validation, P.K., O.B. and Y.T.; formal analysis, Y.T.; investigation, P.K.; resources, O.B.; data curation, Y.T.; writing—original draft preparation, O.B.; writing—review and editing, Y.T.; visualization, P.K.; supervision, P.K.; project administration, Y.T.; funding acquisition, O.B. All authors have read and agreed to the published version of the manuscript.

**Funding:** This research received no external funding.

**Institutional Review Board Statement:** Not applicable.

**Informed Consent Statement:** Not applicable.

**Data Availability Statement:** Data are contained within the article. Information about the influence of substances that were studied in the article is partially presented and is publicly available in «Reports on the assessment of the impact of non-radiation factors on the environment of the Rivne NPP SE "NAEK Energoatom" for the years 2016-2022».

**Conflicts of Interest:** The authors declare no conflict of interest.

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
