# Peer review of "Monitoring of Phosphorus Compounds in the Influence Zone Affected by Nuclear Power Plant Water Discharge in the Styr River (Western Ukraine): Case Study"

_sustainability, doi:10.3390/su152316316_

Round 1

Reviewer 1 Report

Comments and Suggestions for Authors

The case study about “Monitoring of phosphorus compounds in the zone affected by water discharges from a nuclear plant on the Styr River (Western Ukraine)” introduce an effective process to assess the water quality. This research is very interested to ensure the environmental and water safety, and to improve the public health. The experiments and methodology enable the achievement of research objectives. The results are organized and well discussed.  The overall conclusion indicates the absence of a negative impact of phosphorus compounds in the reverse water from Rivne NPP on the ecosystem of the Styr River. However, the relatively high levels of HEDP discharge necessitate the implementation of measures to minimize its use in the technological cycle of Rivne NPP, which could be a perspective direction for further research. Such these researches are valuable for decisions makers. I advise for acceptance in its present form.

Author Response

Dear Reviewer!
We, the Authors of the manuscript, are very grateful for your comments. We tried as much as possible to take all comments into account in the manuscript, so that it became better and more understandable for readers.

Best wishes, Authors

Reviewer 2 Report

Comments and Suggestions for Authors

line 88-89 the term of “small”, “medium”, “large” river are ambiguous. Please clarify the terms.

line 89-90, 155, 174 – directive 2006/44/EC [30] is not longer in force !

change the calculations and perform the analysis in accordance with applicable regulations

Bibliography:

[30] = [40]

The article analyzes the concentration of selected compounds in the river. Only discharges from power plants were considered as the source of compounds. Meanwhile, there may be many sources. Moreover, the concentration may be influenced by the intensification or slowing down of the decomposition of these compounds (as indicated by the authors themselves - lines 288 - 296 and 348 - 352). Not taking these factors into account in the analysis (only the discharge from the power plant) may lead to incorrect conclusions.

note to lines 344 – 356 + table 3

according to the regulations [29] - referred to by the authors - the ecological status of the river is determined by biological, hydromorphological, chemical and physicochemical indicators as well as biological parameters.

The article describes only chemical tests for phosphorus compounds. On what basis do the authors draw the conclusion about the ecological status of the river?

note to lines 361 – 370

C, P and N content tests are not described in the article. On what basis is this analysis made?

note to lines 205 – 210 and 378 – 384

The values resulting from permissible values cannot be mixed with ecological regulations. What is legal does not necessarily have to be ecological.

Author Response

Dear Reviewer!
We, the Authors of the manuscript, are very grateful for your comments. We tried as much as possible to take all comments into account in the manuscript, so that it became better and more understandable for readers.

line 88-89 the term of “small”, “medium”, “large” river are ambiguous. Please clarify the terms.

Done it: line: 103-105

line 89-90, 155, 174 – directive 2006/44/EC [30] is not longer in force !

We did correction

change the calculations and perform the analysis in accordance with applicable regulations

Done

Bibliography:

[30] = [40]

We did correction

The article analyzes the concentration of selected compounds in the river. Only discharges from power plants were considered as the source of compounds. Meanwhile, there may be many sources. Moreover, the concentration may be influenced by the intensification or slowing down of the decomposition of these compounds (as indicated by the authors themselves - lines 288 - 296 and 348 - 352). Not taking these factors into account in the analysis (only the discharge from the power plant) may lead to incorrect conclusions.

Done

note to lines 344 – 356 + table 3

according to the regulations [29] - referred to by the authors - the ecological status of the river is determined by biological, hydromorphological, chemical and physicochemical indicators as well as biological parameters.

The article describes only chemical tests for phosphorus compounds. On what basis do the authors draw the conclusion about the ecological status of the river?

Corrected in the text that the environmental status was determined by chemical indicators of phosphorus compounds, since the authors' task in this case is to show the differences between different approaches to environmental assessment with and without alkalinity. 

note to lines 361 – 370

C, P and N content tests are not described in the article. On what basis is this analysis made?

Line 207-208 The assessment of the biogenic elements was carried out according to the methodology [51], using C:P:N stoichemistry

note to lines 205 – 210 and 378 – 384

The values resulting from permissible values cannot be mixed with ecological regulations. What is legal does not necessarily have to be ecological.

The authors fully share the opinion of the reviewer, but according to Ukrainian standards, the concept of environmental regulations and valuation with values (levels) is established. Therefore, two different methods [34] (Ukraine) and [48] (UKTAG) are presented on page 3.4. The text has been amended to make the different methods clearer.

Once again, we thank you very much for participating in the review of our manuscript!

Best wishes, Authors

Reviewer 3 Report

Comments and Suggestions for Authors

Refers to the manuscript: Monitoring of phosphorus compounds in the zone affected bywater discharges from a nuclear plant on the Styr River (Western Ukraine): Case Study

Thank you for a very interesting case study on monitoring phosphorus compounds in the area of cooling water discharge from a nuclear power plant into the Styr River in Western Ukraine. These studies were designed and conducted correctly. Below are some comments. It would be appropriate to consider their inclusion or clarification in the manuscript. I wish you much success in continuing and expanding your research.

Detailed comments:

1. The title of the manuscript should be corrected to include correct English grammar:

Monitoring of phosphorus compounds in the zone affected by water discharges from a nuclear plant on the Styr River (Western Ukraine): A Case Study

2. The abstract should be more precise and clearly define the goals, concept and methodology of the research. The abstract of a good article in a reputable journal always ends with an outline of the benefits of the results and recommendations as a solution to the problem under study. The presented abstract lacks such information. Please correct and complete the abstract.

3. Keywords should not repeat words contained in the title of the manuscript: „phosphorus compounds”, „discharges”. Please change it.

4. In the Introduction, the authors made an interesting review of the latest literature, but the current state of knowledge regarding the presented problem should be clearly and precisely defined and expanded. In the introduction, a hypothesis should be put forward: how does this work differ from the available literature? The last paragraph concluding the introductory part always emphasizes the innovative aspects of the conducted experiments with clear goals and the importance of the obtained research results for the broadly understood environment. The purpose of the research is not precisely presented and should be formulated in more detail. The manuscript's introduction should be expanded and reformatted to provide a more comprehensive approach.

5. The research methods were correctly selected and described. I have no objections.

6. The results of the research were carefully developed and documented. I have no objections. However, I believe that the authors should compare the obtained results of their own research with the information contained in the literature. Therefore, I propose to supplement the manuscript with a "Discussion" chapter or to combine these two chapters into one "Results and Discussion". All figures and tables presented in the text are legible and correctly interpreted. I have no objections.

7. The Conclusions chapter needs to be slightly improved. They must be persuasive statements about what is considered innovative, with the impact of strongly supporting the results and discussion. It should be clearly noted what new contributions the research has made to the international literature and the final recommendations should be emphasized.

8. Please proofread the manuscript because many sentences are difficult to understand.

Taking into account the comments, I recommend publishing this article.

Comments on the Quality of English Language

Please proofread the manuscript because many sentences are difficult to understand.

Author Response

Dear Reviewer!
We, the Authors of the manuscript, are very grateful for your comments. We tried as much as possible to take all comments into account in the manuscript, so that it became better and more understandable for readers.

Thank you for a very interesting case study on monitoring phosphorus compounds in the area of cooling water discharge from a nuclear power plant into the Styr River in Western Ukraine. These studies were designed and conducted correctly. Below are some comments. It would be appropriate to consider their inclusion or clarification in the manuscript. I wish you much success in continuing and expanding your research.

Detailed comments:

  1. The title of the manuscript should be corrected to include correct English grammar:

Monitoring of phosphorus compounds in the zone affected by water discharges from a nuclear plant on the Styr River (Western Ukraine): A Case Study

We done it.

  1. The abstract should be more precise and clearly define the goals, concept and methodology of the research. The abstract of a good article in a reputable journal always ends with an outline of the benefits of the results and recommendations as a solution to the problem under study. The presented abstract lacks such information. Please correct and complete the abstract.
  2. Keywords should not repeat words contained in the title of the manuscript: „phosphorus compounds”, „discharges”. Please change it.

We done it.

  1. In the Introduction, the authors made an interesting review of the latest literature, but the current state of knowledge regarding the presented problem should be clearly and precisely defined and expanded. In the introduction, a hypothesis should be put forward: how does this work differ from the available literature? The last paragraph concluding the introductory part always emphasizes the innovative aspects of the conducted experiments with clear goals and the importance of the obtained research results for the broadly understood environment. The purpose of the research is not precisely presented and should be formulated in more detail. The manuscript's introduction should be expanded and reformatted to provide a more comprehensive approach.

We done it.

  1. The research methods were correctly selected and described. I have no objections.
  2. The results of the research were carefully developed and documented. I have no objections. However, I believe that the authors should compare the obtained results of their own research with the information contained in the literature. Therefore, I propose to supplement the manuscript with a "Discussion" chapter or to combine these two chapters into one "Results and Discussion". All figures and tables presented in the text are legible and correctly interpreted. I have no objections.
  3. The Conclusions chapter needs to be slightly improved. They must be persuasive statements about what is considered innovative, with the impact of strongly supporting the results and discussion. It should be clearly noted what new contributions the research has made to the international literature and the final recommendations should be emphasized.
  4. Please proofread the manuscript because many sentences are difficult to understand.

We tried to shorten the sentences so that the text was more accessible to readers

Taking into account the comments, I recommend publishing this article.

Once again, we thank you very much for participating in the review of our manuscript!

Best wishes, Authors

Round 2

Reviewer 2 Report

Comments and Suggestions for Authors

Changes suggested by all reviewers has been made.

Author Response

Dear reviewer, thank you for your fruitful contribution to the improvement of our article.

with best regards, Authors